# RiboSphere: Learning Unified and Efficient Representations of RNA Structures

**Zhou Zhang** [* 1]  **Hanqun Cao** [* 2]  **Cheng Tan** [† 2]  **Fang Wu** [† 3]  **Pheng Ann Heng** [2]  **Tianfan Fu** [† 1]

## Abstract

Accurate RNA structure modeling remains difficult because RNA backbones are highly flexible, non-canonical interactions are prevalent, and experimentally determined 3D structures are comparatively scarce. We introduce *RiboSphere*, a framework that learns *discrete* geometric representations of RNA by combining vector quantization with flow matching. Our design is motivated by the modular organization of RNA architecture: complex folds are composed from recurring structural motifs. RiboSphere uses a geometric transformer encoder to produce approximately SE(3)-invariant (rotation/translation-invariant) features, which are discretized with finite scalar quantization (FSQ) into a finite vocabulary of latent codes. Conditioned on these discrete codes, a flow-matching decoder reconstructs atomic coordinates, enabling high-fidelity structure generation. We find that the learned code indices are enriched for specific RNA motifs, suggesting that the model captures motif-level compositional structure rather than acting as a purely compressive bottleneck. Across benchmarks, RiboSphere achieves strong performance in structure reconstruction (RMSD 1.25 Å, TM-score 0.84), and its pretrained discrete representations transfer effectively to inverse folding and RNA–ligand binding prediction, with robust generalization in data-scarce regimes. Code is available here: https://github.com/Zhangz312/RiboSphere

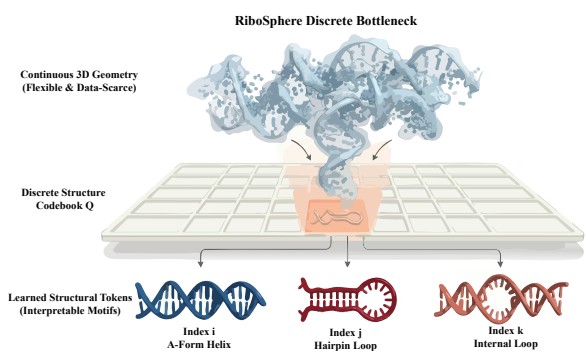

*Figure 1.* **Overview of RiboSphere**: from continuous RNA geometry to discrete, interpretable structural units.

---

[*]Equal contribution  [†]Corresponding authors  [1]State Key Laboratory for Novel Software Technology at Nanjing University, School of Computer Science, Nanjing University [2]The Chinese University of Hong Kong [3]Stanford University. Correspondence to: Cheng Tan <chengtan@cuhk.edu.hk>, Fang Wu <fangwu97@stanford.edu>, Tianfan Fu <futianfan@nju.edu.cn>.

*Proceedings of the 43rd International Conference on Machine Learning*, Seoul, South Korea. PMLR 306, 2026. Copyright 2026 by the author(s).

## 1. Introduction

Ribonucleic acid (RNA) plays multifaceted roles in living systems, ranging from genetic intermediaries to functional regulators (Goodall & Wickramasinghe, 2021; Morris & Mattick, 2014), with its diverse biological functions intimately linked to its complex three-dimensional conformations (Crick, 1970). Developing a universal RNA structural encoder capable of capturing intricate geometric constraints while maintaining robustness in data-scarce regimes has emerged as a central challenge at the intersection of computational biology and artificial intelligence (Townshend et al., 2020). Despite the remarkable success of models such as AlphaFold in the protein domain (Jumper et al., 2021; Abramson et al., 2024), RNA structure modeling continues to face formidable obstacles (Bernard et al., 2024; Martinović et al., 2024).

Current methods for RNA structure representation face three key challenges: **(1) Data scarcity.** While the Protein Data Bank contains over 200,000 protein structures, fewer than 6,000 RNA 3D structures have been experimentally resolved (Adamczyk et al., 2022). This order-of-magnitude gap renders conventional end-to-end representation learning prone to overfitting, and transfer learning from proteins offers limited remedy given fundamental differences in backbone chemistry. **(2) Continuous space limitations.** Existing generative approaches operating in continuous latent spaces face challenges on sparse data manifolds (Ramakers et al., 2024): VAEs suffer from posterior collapse with expressive decoders, while diffusion models may produce biophysically

implausible conformations without strong geometric priors. The high backbone flexibility of RNA further exacerbates these issues. **(3) Lack of interpretability.** Most learned representations function as black-box embeddings, lacking correspondence to biologically meaningful units. Yet RNA conformations are composed of recurring motifs—hairpin loops, internal loops, junctions—that serve as functional building blocks (Leontis et al., 2006). Capturing this modularity would improve both generalization and mechanistic understanding.

These challenges highlight a core bottleneck of existing RNA representations: continuous latent spaces struggle to generalize under data scarcity and fail to capture the modular, motif-based organization of RNA structures. As illustrated in Figure 1, this motivates a shift from continuous RNA geometry toward discrete and interpretable structural units.

To address these challenges, we propose RiboSphere, a framework that learns discrete geometric representations of RNA by integrating Vector Quantization (VQ) (Van Den Oord et al., 2017) with Flow Matching (FM) (Lipman et al., 2023). Our key insight is that complex RNA conformations are not amorphous continua but rather compositional assemblies of a finite set of geometrically distinctive structural motifs, such as hairpin loops, internal loops, and pseudoknots (Leontis et al., 2006; Duarte et al., 2003). RiboSphere leverages this modularity by projecting local geometries onto a learnable discrete codebook via Finite Scalar Quantization (FSQ), while a flow-based decoder reconstructs atomic coordinates from these discrete priors with sub-angstrom fidelity.

Our main contributions are three-fold:

- We propose RiboSphere, a VQ-Flow framework that combines a geometric transformer encoder with FSQ discretization and flow matching decoding. The FSQ bottleneck naturally avoids posterior collapse without auxiliary losses, and the asymmetric architecture (shallow encoder, deep decoder) encourages compression of key structural features into the discrete latent space.

- We demonstrate that the learned codebook captures biologically meaningful structure: token sequences exhibit high geometric consistency (RMSD $< 0.5$ Å), and different RNA motifs (hairpin loops, internal loops, three-way junctions) show statistically distinct token distributions, suggesting the model learns interpretable structural primitives rather than arbitrary compression.

- Extensive experiments show that RiboSphere achieves state-of-the-art structure reconstruction (RMSD 1.25 Å, TM-score 0.84), the highest sequence recovery in inverse folding (63.0%), and strong generalization

in RNA-ligand binding prediction under out-of-distribution splits, outperforming GerNA-Bind by 2.6% on the most challenging homology-fingerprint split.

## 2. Related Work

### 2.1. RNA Structure Representation

RNA 3D structure representation has evolved from computer vision-inspired volumetric grids to physics-aware geometric graphs. Early approaches leveraged 3D CNNs by discretizing molecular space into voxels to capture atomic densities (Li et al., 2018; Peng et al., 2020; Sun & Gao, 2024). While these grid-based methods pioneered quality assessment and binding site detection, they suffer from sparsity and lack rotational invariance, requiring expensive data augmentation (Das et al., 2010; Li et al., 2019). Alternative work explored backbone torsion angles to discretize folding into "structural alphabets" (Hershkovitz et al., 2003), but these internal coordinates suffer from the "lever-arm" effect, where minor angular errors propagate into large global deviations.

The current frontier has shifted toward E(3)-equivariant Graph Neural Networks. By encoding physical symmetries directly into the architecture, models like ARES (Townshend et al., 2021) achieve remarkable data efficiency, outperforming voxel-based methods even with minimal training samples. Recent frameworks including EquiRNA (Li et al., 2025) and gRNAde (Joshi et al., 2025) extend this paradigm through hierarchical representations and multi-state GNNs, enabling modeling of large-scale complexes and dynamic conformational ensembles. We build upon these geometric foundations with a discrete latent space that preserves symmetries while enabling efficient generative sampling.

### 2.2. Autoencoders for Biomolecules

Generative modeling for biomolecules has transitioned from continuous latent models to discrete tokenization frameworks that better align with the hierarchical nature of biological structures. Early research utilized VAEs with continuous Gaussian priors to capture evolutionary constraints (Riesselman et al., 2018), but these models often suffer from posterior collapse when paired with expressive decoders and struggle to resolve the rugged energy structure landscapes (Ramakers et al., 2024). VQ-VAEs (Xue et al., 2019) emerged as a robust alternative, discretizing the latent space into a codebook of structural motifs. This paradigm has been successfully applied to protein engineering: FoldToken (Gao et al., 2025) uses soft quantization to enable sub-Angstrom backbone reconstruction, while GCP-VQVAE (Pourmirzaei et al., 2025) and ESM3 (Hayes et al., 2025) integrate SE(3)-equivariant encoders to keep geometric consistency. Hybrid approaches like ProVQ (Liu et al., 2025b) and SLM (Lu et al., 2025) further combine VQ-VAEs with

diffusion models to capture dynamic conformational ensembles.

In the RNA domain, generative modeling is still navigating the balance between sequence-based fluency and structural fidelity. While GenerRNA (Zhao et al., 2024) leverages large-scale language modeling to learn RNA "grammar" from sequences, structural models like Dfold (Ramakers et al., 2024) have only recently introduced VQ-VAEs for de novo 3D prediction. However, current RNA-specific models often rely on coarse-grained representations, such as three-class distance binning, which lack the resolution to model intricate non-canonical interactions. Furthermore, existing methods primarily focus on single-chain RNA folding, leaving a methodological gap in modeling multi-chain interfaces. While RiboSphere is currently trained and evaluated on single-chain RNA, the underlying discrete, geometry-complete quantization framework is conceptually compatible with multi-chain docking and interface modeling, which we plan to explore in future work.

## 3. Method

### 3.1. Preliminaries

We address the problem of learning representations and generating three-dimensional structures of RNA molecules. An RNA molecule is defined by its sequence $\mathbf{s} = (s_1, s_2, \ldots, s_L)$ where $s_i \in \{A, U, C, G\}$ denotes the nucleotide type, and its three-dimensional structure represented as atomic coordinates:

$$\mathbf{x} \in \mathbb{R}^{L \times A \times 3}, \tag{1}$$

where $L$ is the sequence length and $A$ is the number of atoms per nucleotide. Depending on the modeling granularity, we define three atomic-level representation strategies:

- **Single-atom model**: $A = 1$, using only the C4' atom.

- **10-atom model**: $A = 10$, with the backbone atom set $\{P, C5', C4', C3', C2', C1', O5', O4', O3', O2'\}$.

- **11-atom model**: $A = 11$, extending the 10-atom backbone with a base-anchoring atom: N9 for purines or N1 for pyrimidines.

All input coordinates are mean-centered and augmented with random rotations during training to learn approximately SE(3)-invariant representations.

**Learning Objectives.** Our framework employs a two-stage learning paradigm. In the *pretraining stage*, we learn a discrete geometric encoder via structure reconstruction, formulated as maximizing the likelihood of atomic coordinates given the discrete latent representation:

$$\mathcal{L}_{\text{recon}} = -\mathbb{E}_{\mathbf{x} \sim p_{\text{data}}} \left[ \log p_\phi(\mathbf{x} \mid \hat{\mathbf{c}}) \right], \quad \hat{\mathbf{c}} = \mathcal{Q}(\mathcal{E}_\theta(\mathbf{x})), \tag{2}$$

where $\mathcal{E}_\theta$ denotes the encoder, $\mathcal{Q}$ the vector quantization module, and $p_\phi$ the flow-based decoder distribution.

In the *downstream tasks*, including inverse folding and RNA-ligand binding, we freeze the pretrained encoder and quantizer, transferring the learned discrete representations $\hat{\mathbf{c}}$ and continuous representations $\mathcal{E}_\theta(\mathbf{x})$ to task-specific objectives:

### 3.2. Encoder Architecture and Featurization

As illustrated in Figure 2(a), the encoder $\mathcal{E}_\theta$ maps RNA atomic coordinates $\mathbf{x} \in \mathbb{R}^{L \times A \times 3}$ to a latent sequence $\mathbf{c} = (c_1, c_2, \ldots, c_L)$ with $c_i \in \mathbb{R}^d$:

$$\mathbf{c} = \mathcal{E}_\theta(\mathbf{x}). \tag{3}$$

We first apply mean-centering to obtain $\tilde{\mathbf{x}}$, then flatten the atomic coordinates of each nucleotide and project them to the encoder hidden dimension $d$:

$$h_i = \text{MLP}_{\text{in}}(\text{vec}(\tilde{\mathbf{x}}_i)), \quad h_i \in \mathbb{R}^d. \tag{4}$$

To capture inter-nucleotide spatial relationships and sequence topology, we construct pairwise features by combining discretized distance embeddings and relative positional encodings:

$$p_{ij} = \text{MLP}_{\text{pair}}(\text{LN}(\mathbf{e}_{ij}^{\text{dist}} + \mathbf{e}_{ij}^{\text{pos}})), \tag{5}$$

where $\mathbf{e}_{ij}^{\text{dist}}$ is obtained by bucketing pairwise Euclidean distances and embedding the resulting indices, and $\mathbf{e}_{ij}^{\text{pos}}$ encodes truncated relative sequence positions.

The encoder employs multi-head self-attention, injecting pairwise features $p_{ij}$ as geometric biases:

$$\text{Att}(h_i, h_j) = \text{softmax} \left( \frac{(h_i \mathbf{W}_Q)(h_j \mathbf{W}_K)^\top}{\sqrt{d_k}} + p_{ij} \right) h_j \mathbf{W}_V, \tag{6}$$

where $\mathbf{W}_Q, \mathbf{W}_K, \mathbf{W}_V$ are learnable projection matrices and $d_k$ is the attention head dimension. Through multi-layer stacking combined with a sliding window mechanism, the encoder effectively models both global and local spatial dependencies, yielding context-enriched representations:

$$c_i = h_i + \sum_{j \neq i} \text{Att}(h_i, h_j), \quad i = 1, \ldots, L. \tag{7}$$

### 3.3. Vector Quantization Bottleneck

The encoder output $\mathbf{c}$ is discretized via Finite Scalar Quantization (FSQ) (Mentzer et al., 2023):

$$\hat{c} = \lfloor l/2 \rfloor \cdot \tanh(\text{Linear}(\mathbf{c})), \tag{8}$$

where $l$ denotes the number of quantization levels per dimension. The quantized $\hat{\mathbf{c}}$ serves as a discrete geometric summary shared across all downstream tasks, with the straight-through estimator employed during training to enable gradient backpropagation.

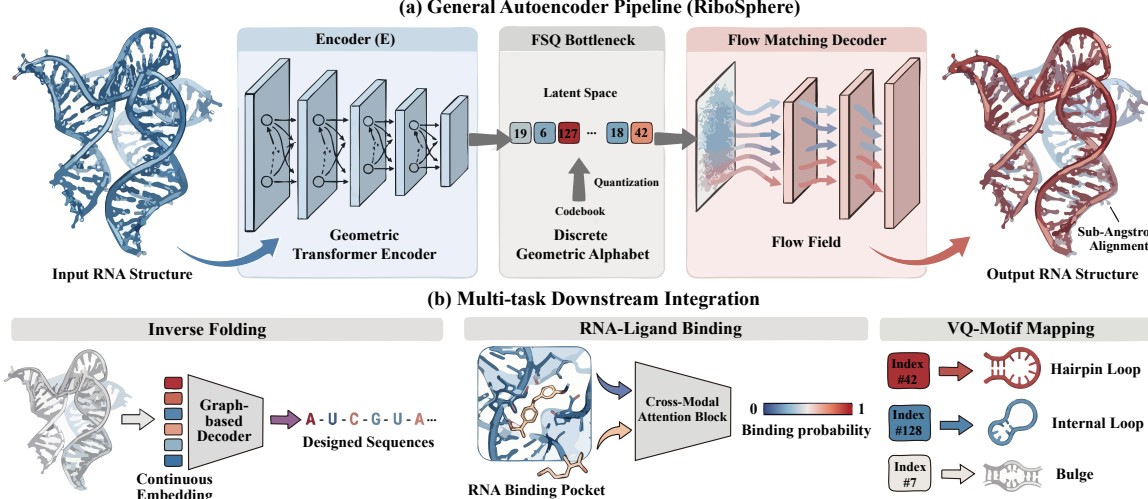

*Figure 2.* Overall pipeline of **RiboSphere**. (a) **General autoencoder pipeline.** RNA atomic coordinates are encoded by a geometric transformer into continuous latent representations, which are discretized via Finite Scalar Quantization (FSQ) to obtain discrete geometric tokens. A flow-matching decoder reconstructs full 3D structures from the discrete representations, enabling high-fidelity structure reconstruction and serving as the pretraining objective. (b) **Multi-task downstream integration.** The pretrained encoder and quantizer are frozen and reused across downstream tasks. Discrete structural tokens and continuous embeddings are transferred to task-specific architectures for inverse folding and RNA–ligand binding prediction, providing a shared and interpretable geometric backbone.

## 3.4. Decoder and Structural Reconstruction

We adopt Flow Matching (Lipman et al., 2023) for continuous generative modeling of 3D structures. Given randomly initialized noise conformations $\mathbf{x}_t \in \mathbb{R}^{L \times A \times 3}$ and discrete conditioning $\hat{\mathbf{c}}$, the decoder learns a time-dependent vector field:

$$\mathcal{D}_\phi(\mathbf{x}_t, t, \hat{\mathbf{c}}) = v_\theta(\mathbf{x}_t, t, \hat{\mathbf{c}}), \quad t \in [0, 1]. \qquad (9)$$

The flow matching formulation enables flexible incorporation of advanced sampling strategies at inference time. Euler integration yields the sampling trajectory:

$$\mathbf{x}_{t+\Delta t} = \mathbf{x}_t + v_\theta(\mathbf{x}_t, t, \hat{\mathbf{c}})\Delta t, \qquad (10)$$
$$\text{where} \quad t \in \left\{ 0, \frac{1}{N}, \frac{2}{N}, \dots, \frac{N-1}{N} \right\}.$$

Classifier-free guidance (Ho & Salimans, 2022) further enhances generation quality by amplifying sensitivity to conditioning information:

$$\tilde{v}_\theta(\mathbf{x}_t, t, \hat{\mathbf{c}}) = v_\theta(\mathbf{x}_t, t, \hat{\mathbf{c}}) + g \cdot (v_\theta(\mathbf{x}_t, t, \hat{\mathbf{c}}) - v_\theta(\mathbf{x}_t, t, \varnothing)), \qquad (11)$$

where $g$ is the guidance weight and $\varnothing$ denotes the null conditioning vector.

For Gaussian flow, the vector field admits an analytical conversion to the score field:

$$s_\theta(\mathbf{x}_t, t, \hat{\mathbf{c}}) = \frac{t \cdot v_\theta(\mathbf{x}_t, t, \hat{\mathbf{c}}) - \mathbf{x}_t}{1 - t}, \qquad (12)$$

which enables stochastic differential equation (SDE) sampling for enhanced generation diversity:

$$d\mathbf{x}_t = v_\theta(\mathbf{x}_t, t, \hat{\mathbf{c}})dt + g(t)\eta \cdot s_\theta(\mathbf{x}_t, t, \hat{\mathbf{c}})dt + \sqrt{2g(t)\gamma}\, d\mathbf{W}_t, \qquad (13)$$

where $\eta$ and $\gamma$ control the score gradient scaling and noise intensity, respectively.

The model is trained by minimizing the flow matching loss:

$$\mathcal{L}_{\text{flow}} = \mathbb{E}_{\mathbf{x}_0, \mathbf{x}_1, t}\left[ \|v_\theta(\mathbf{x}_t, t, \hat{\mathbf{c}}) - (\mathbf{x}_1 - \mathbf{x}_0)\|_2^2 \right]. \qquad (14)$$

## 3.5. Inverse Folding

We adapt discrete structural representations for the generation of sequences conditioned by structure, following the geometric inverse folding paradigm (Joshi et al., 2025).

**Backbone Encoding.** For inverse folding, we employ a 6-atom coarse-grained representation retaining the coordinates $\{P, C5', C4', C3', O5', O3'\}$ for each nucleotide. The frozen encoder maps the backbone to discrete structural features $\hat{\mathbf{c}} \in \mathbb{R}^{L \times d}$, which capture SE(3)-invariant geometric patterns through rotation augmentation during pretraining.

**Geometry Adapter.** To supplement directional geometric relationships beyond scalar discrete features, we introduce a Geometry Adapter that constructs rotation-equivariant representations. For each nucleotide $i$, we compute local coordinate frames from backbone atoms and extract directional vectors to form a joint scalar-vector representation:

$$h_i^{\text{geo}} = (s_i, \mathbf{v}_i), \quad s_i \in \mathbb{R}^{d_s}, \quad \mathbf{v}_i \in \mathbb{R}^{K \times 3} \qquad (15)$$

*Table 1.* Comparison of RNA Structure Reconstruction and Quantization Methods.

| Method | # Enc | # Dec | Dim | # Atoms | Structure | | | VQ | |
|---|---|---|---|---|---|---|---|---|---|
| | | | | | RMSD | TM-score | lDDT | Codebook | % Util. |
| E2-D8, D256, A1 | 2 | 8 | 256 | 1 | 2.14 | 0.70 | 0.73 | 240 | 100.0 |
| E2-D8, D256, A10 | 2 | 8 | 256 | 10 | 1.80 | 0.76 | 0.77 | 240 | 100.0 |
| E2-D8, D256, A11 | 2 | 8 | 256 | 11 | 2.05 | 0.71 | 0.73 | 240 | 100.0 |
| E6-D6, D512, A1 | 6 | 6 | 512 | 1 | 1.88 | 0.73 | 0.75 | 1,000 | 95.0 |
| E6-D6, D512, A10 | 6 | 6 | 512 | 10 | 2.44 | 0.67 | 0.71 | 1,000 | 88.5 |
| E6-D6, D512, A11 | 6 | 6 | 512 | 11 | 2.71 | 0.68 | 0.72 | 1,000 | 98.9 |
| E2-D8, D256, A1 | 2 | 8 | 256 | 1 | 1.88 | 0.75 | 0.76 | 1,000 | 80.7 |
| E2-D8, D256, A10 | 2 | 8 | 256 | 10 | 2.33 | 0.69 | 0.74 | 1,000 | 84.2 |
| E2-D8, D256, A11 | 2 | 8 | 256 | 11 | 1.60 | 0.78 | 0.79 | 1,000 | 88.0 |
| E2-D8, D256, A1 | 2 | 8 | 256 | 1 | 1.25 | 0.84 | 0.83 | 4,375 | 39.8 |
| E2-D8, D256, A10 | 2 | 8 | 256 | 10 | 1.58 | 0.80 | 0.76 | 4,375 | 39.4 |
| E2-D8, D256, A11 | 2 | 8 | 256 | 11 | 1.35 | 0.82 | 0.82 | 4,375 | 39.9 |
| E4-D8, D256, A11 | 4 | 8 | 256 | 11 | 2.26 | 0.69 | 0.73 | 240 | 100.0 |
| E4-D6, D256, A11 | 4 | 6 | 256 | 11 | 2.43 | 0.67 | 0.72 | 240 | 100.0 |
| E6-D4, D256, A11 | 6 | 4 | 256 | 11 | 2.93 | 0.60 | 0.67 | 240 | 100.0 |
| E8-D2, D256, A11 | 8 | 2 | 256 | 11 | 3.77 | 0.53 | 0.62 | 240 | 100.0 |

where $s_i$ denotes rotation-invariant scalar features combining $\hat{c}_i$ with secondary structure and base-pairing priors, and $\mathbf{v}_i$ represents rotation-equivariant vector features derived from local frame orientations.

**Autoregressive Decoding.** The sequence decoder employs multi-layer Geometric Vector Perceptrons (GVP) (Jing et al., 2021) for autoregressive prediction from the 5' to 3' end. At each position $i$, the decoder predicts:

$$p_\psi(s_i \mid \hat{\mathbf{c}}, s_{<i}) = \text{softmax}\left(\text{GVP}_{\text{dec}}(h_i^{\text{geo}}, h_{<i}^{\text{seq}})\right) \quad (16)$$

where $h_{<i}^{\text{seq}}$ encodes the partially decoded sequence context.

**Training Objective.** The model is optimized via cross-entropy loss with label smoothing ($\epsilon = 0.1$):

$$\mathcal{L}_{\text{inv}} = -\sum_{i=1}^{L}(1-\epsilon)\log p_\psi(s_i \mid \hat{\mathbf{c}}, s_{<i})$$
$$+ \frac{\epsilon}{4}\sum_{s' \in \mathcal{V}}\log p_\psi(s' \mid \hat{\mathbf{c}}, s_{<i})], \quad (17)$$

where $\mathcal{V} = \{\text{A}, \text{U}, \text{C}, \text{G}\}$ is the nucleotide vocabulary.

### 3.6. RNA-Ligand Binding Prediction

We integrate the discrete structural representations into the GerNA-Bind framework (Xia et al., 2025) for RNA-ligand binding prediction.

**RNA Encoding.** The RNA is represented through multi-modal features. Our pretrained discrete structural embedding $\hat{\mathbf{c}}$ serves as the 3D geometric backbone, combined with:

$$\mathbf{h}_{\text{RNA}} = \text{MLP}_{\text{fuse}}\left(\hat{\mathbf{c}}\|\mathbf{e}^{\text{1D}}\|\mathbf{e}^{\text{2D}}\right) \in \mathbb{R}^{L \times d} \quad (18)$$

where $\mathbf{e}^{\text{1D}}$ denotes pretrained sequence embeddings and $\mathbf{e}^{\text{2D}}$ encodes secondary structure graph features.

**Ligand Encoding.** The ligand is encoded through both 2D molecular graph and 3D conformational information:

$$\mathbf{h}_{\text{lig}} = \text{GraphTransformer}(\mathbf{G}_{\text{mol}}, \mathbf{X}_{\text{mol}}) \in \mathbb{R}^{N \times d} \quad (19)$$

where $\mathbf{G}_{\text{mol}}$ is the molecular graph and $\mathbf{X}_{\text{mol}} \in \mathbb{R}^{N \times 3}$ contains 3D atomic coordinates.

**Interaction Modeling.** RNA-ligand interactions are modeled through a pairwise feature tensor refined by triangular attention:

$$\mathbf{Z}_{ij}^{(0)} = \text{MLP}\left(\mathbf{h}_{\text{RNA},i}\|\mathbf{h}_{\text{lig},j}\|d_{ij}\right) \in \mathbb{R}^{d_p}, \quad (20)$$

$$\mathbf{Z}_{ij}^{(\ell+1)} = \mathbf{Z}_{ij}^{(\ell)} + \text{TriangleAttn}\left(\mathbf{Z}^{(\ell)}, \mathbf{h}_{\text{RNA}}, \mathbf{h}_{\text{lig}}\right), \quad (21)$$

where $d_{ij}$ encodes the spatial distance between RNA residue $i$ and ligand atom $j$.

**Affinity Prediction.** The final binding affinity is predicted via cross-modal attention pooling:

$$\mathbf{z}_{\text{global}} = \text{CrossAttnPool}(\mathbf{Z}^{(L)}, \mathbf{h}_{\text{RNA}}, \mathbf{h}_{\text{lig}}), \quad (22)$$

$$\hat{y} = \sigma\left(\text{MLP}_{\text{pred}}(\mathbf{z}_{\text{global}})\right), \quad (23)$$

where $\sigma$ is the sigmoid function.

**Training Objective.** Following GerNA-Bind (Xia et al., 2025), we adopt an evidential deep learning objective:

$$\mathcal{L}_{\text{bind}} = \sum_i \left[ (y_i - \hat{p}_i)^2 + \frac{\hat{p}_i(1 - \hat{p}_i)}{S_i + 1} \right] \\ + \lambda_t \cdot \text{KL} \left[ \text{Dir}(\tilde{\boldsymbol{\alpha}}_i) \| \text{Dir}(\mathbf{1}) \right], \tag{24}$$

where $\hat{p}_i$ and $S_i$ are derived from a Dirichlet-parameterized output, and $\lambda_t$ is an annealing coefficient.

# 4. Experiments

We evaluate RiboSphere across a series of tasks to assess the quality, generalization, and interpretability of its discrete RNA structural representations. Section 4.1 introduces the experimental settings and evaluation metrics. Section 4.2 studies the reconstruction of the RNA structure to analyze how the design of the codebook and the architectural choices affect the geometric expressivity. Section 4.3 evaluates the representations in RNA inverse folding from a generative perspective, while Section 4.4 examines their effectiveness in RNA–ligand binding prediction, particularly under out-of-distribution splits. Section 4.5 further analyzes the VQ codebook to validate the biological relevance and interpretability of the learned discrete structural tokens.

## 4.1. Experimental Settings

### Dataset.

**(i) Reconstruction**: The reconstruction task adopts the single-state split setting from gRNAde. Specifically, following the RNA structural clusters identified by Das et al. (Das et al., 2010), all RNA samples belonging to these clusters are collectively assigned to the test set, ensuring a strict structural decoupling between the test set and the training data. The remaining structural clusters that are not selected for the test set are randomly split into the training and validation sets. To further increase the effective sample size and enhance structural diversity, we explicitly expand multiple conformations associated with each RNA sequence, treating each conformation as an independent training instance. After this multi-conformation expansion, the resulting dataset consists of 11,183 training samples, 551 validation samples, and 239 test samples.

**(ii) Inverse folding**: The dataset follows gRNAde (Joshi et al., 2025), which is derived from RNASolo (Adamczyk et al., 2022), containing 4,223 unique RNA sequences and 12,011 structures. The dataset is split into single-state and multi-state clusters based on RNA structural flexibility.

**(iii) RNA-ligand binding**: The Robin dataset, derived from high-throughput microarray screening, focuses on 26 RNA targets and 1,893 drug-like molecules to provide 46,052 filtered interactions. Complementing this, the Biosensor dataset includes 191 synthetic ribozyme-aptamer RNAs and 89 drug-like molecules, totaling 16,999 interactions for evaluation. Rigorous evaluation is ensured through four split strategies: random, RNA homology-based, ligand fingerprint-based, and a combined homology-fingerprint approach (Xia et al., 2025).

**Metrics. Reconstruction**: The reconstruction task is primarily evaluated from the perspective of three-dimensional structural similarity. Performance is assessed using Root Mean Square Deviation (RMSD) to measure the average deviation between predicted and true atomic coordinates, TM-score to evaluate the similarity of the overall structural topology, and the local Distance Difference Test (lDDT) to quantify the accuracy of local geometric relationships in the predicted structures.

**Inverse folding**: Performance is evaluated using sequence metrics (recovery and diversity), 2D structural metrics from EternaFold (scMCC), and 3D structural metrics from Rho-Fold (RMSD, TM-score, pLDDT).

**RNA-ligand binding**: Binding specificity is measured by area under the receiver operating characteristic (AUROC).

## 4.2. RNA Structure Reconstruction

We evaluated RiboSphere under various configurations to identify the core design factors affecting reconstruction performance. As shown in Table 1, three key findings emerge:

**Atomic granularity matters, but requires sufficient codebook capacity.** Single-atom (C4′) representations preserve overall topology but lack base orientation information, introducing systematic errors at the atomic level. Adopting 10- or 11-atom representations enables the model to capture backbone-base configurations explicitly, yielding more accurate reconstruction. However, this benefit materializes only when the codebook is large enough to support the increased geometric diversity.

**Larger codebooks enable selective, semantically consistent encoding.** With a small codebook (240 tokens), the model achieves 100% utilization but forces distinct conformations onto shared tokens, losing local geometric detail. As codebook size increases to 4,375, a sparse encoding pattern emerges spontaneously: utilization drops to ∼40%, yet reconstruction improves. Rather than covering all tokens uniformly, RiboSphere selects a discriminative subset that captures semantically consistent local structures. This selective usage prevents posterior collapse while enhancing interpretability.

**Asymmetric architecture outperforms deeper encoders.** Contrary to intuition, performance does not increase monotonically with encoder depth. The best results arise from a shallow encoder (2 layers) paired with a deep decoder (8

*Table 2.* Performance Metrics for Protein Sequence Prediction (**best**, second best).

| Method | Sequence | | 2D Struct. | 3D Struct. | | |
|---|---|---|---|---|---|---|
| | Div. (↑) | Rec. (↑) | scMCC (↑) | RMSD (↓) | TM-score (↑) | pLDDT (↑) |
| RiFold (Liu et al., 2025a) | **0.89** | 0.416 | 0.28 | 17.06 | 0.12 | 0.50 |
| RDesign (Tan et al., 2024) | 0.84 | 0.415 | 0.20 | 16.81 | 0.12 | 0.45 |
| RIDiffusion (Hou et al., 2025) | 0.85 | 0.533 | 0.59 | **10.66** | 0.25 | 0.60 |
| gRNAde (Joshi et al., 2025) | 0.83 | 0.529 | 0.60 | 11.45 | **0.29** | **0.62** |
| RiboSphere | 0.82 | **0.630** | **0.63** | 11.54 | **0.29** | 0.55 |

*Table 3.* Comparison of AUROC Metrics across Biosensor and Robin Datasets (**best**, second best).

| Method | AUROC-Biosensor | | | | AUROC-Robin | | | |
|---|---|---|---|---|---|---|---|---|
| | Random | RNA homology | Ligand fingerprint | Homol. & fingerpr. | Random | RNA homology | Ligand fingerprint | Homol. & fingerpr. |
| RSAPred (Krishnan et al., 2024) | 0.8764 | 0.7550 | 0.6707 | 0.6019 | 0.6327 | 0.5392 | 0.6320 | 0.4938 |
| DeepDTIs (Wen et al., 2017) | 0.9300 | 0.8399 | 0.6813 | 0.6118 | 0.6302 | 0.5503 | 0.6290 | 0.4987 |
| DeepConv-DTI (Lee et al., 2019) | 0.9249 | 0.8427 | 0.6894 | 0.6213 | 0.6301 | 0.5625 | 0.6390 | 0.5104 |
| GraphDTA (Nguyen et al., 2021) | 0.8992 | 0.8284 | 0.7014 | 0.6370 | 0.6590 | 0.5528 | 0.6481 | 0.5510 |
| GerNA-Bind (Xia et al., 2025) | **0.9755** | 0.9014 | **0.7723** | 0.7279 | **0.7094** | 0.6188 | 0.6814 | 0.6176 |
| RiboSphere | 0.9524 | **0.9031** | 0.7406 | **0.7534** | 0.6945 | **0.6335** | **0.6950** | **0.6349** |

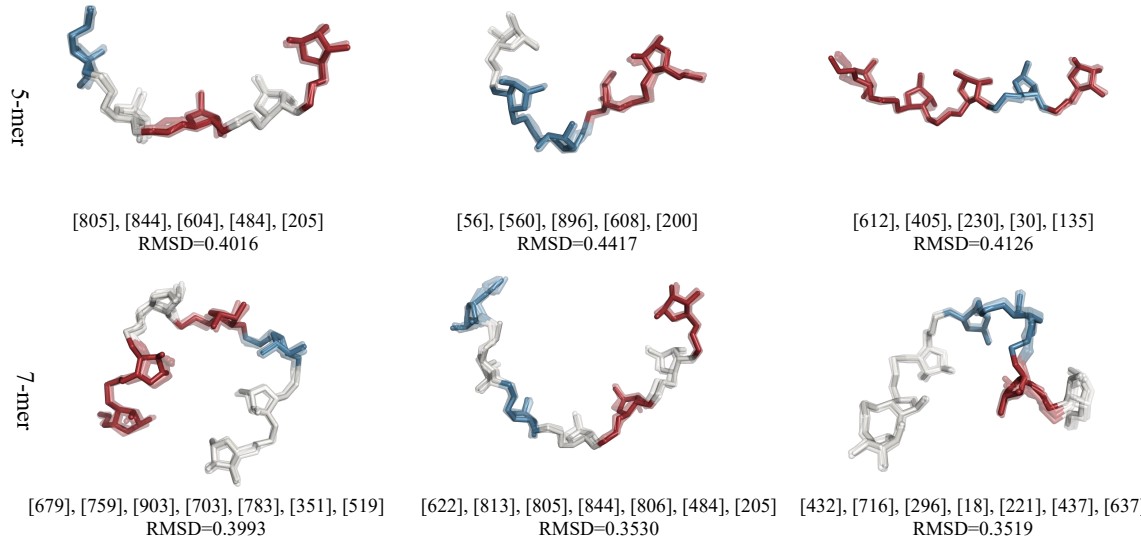

5-mer

[805], [844], [604], [484], [205]
RMSD=0.4016

[56], [560], [896], [608], [200]
RMSD=0.4417

[612], [405], [230], [30], [135]
RMSD=0.4126

7-mer

[679], [759], [903], [703], [783], [351], [519]
RMSD=0.3993

[622], [813], [805], [844], [806], [484], [205]
RMSD=0.3530

[432], [716], [296], [18], [221], [437], [637]
RMSD=0.3519

*Figure 3.* Structural consistency of high-frequency VQ token sequences.

layers). Overly expressive encoders can bypass the discrete bottleneck via complex continuous mappings, weakening the structural constraints imposed by quantization. A compact encoder forces key geometric features into the discrete latent space, improving representation robustness.

## 4.3. Inverse Folding

We apply RiboSphere's structural embeddings to RNA inverse folding. As shown in Table 2, RiboSphere achieves a sequence recovery rate of 63.0%, substantially outperforming baselines including gRNAde (52.9%) and RIDiffusion (53.3%). Sequence diversity is slightly lower (0.82 vs. 0.89

for RiFold), reflecting a design choice: our model prioritizes structural fidelity over maximal diversity, which is often the primary objective in inverse folding.

For 2D self-consistency, designed sequences were folded with EternaFold. RiboSphere achieves the highest scMCC (0.633), indicating that its sequences best preserve secondary structure—consistent with the VQ bottleneck capturing local motifs. For 3D self-consistency, folded structures from RhoFold yield RMSD (11.54), TM-score (0.290), and pLDDT (0.548) comparable to gRNAde, while slightly lower pLDDT reflects predictor limitations. Overall, RiboSphere maintains strong sequence-structure consistency, particularly on structurally complex targets where the discrete codebook provides informative priors for non-canonical regions such as internal loops and junctions.

### 4.4. RNA-Ligand Binding

We evaluated RiboSphere on RNA-ligand binding prediction across two datasets (Biosensor and Robin) with four splitting strategies of increasing difficulty (Table 3). RiboSphere achieves the best performance on 5 of 8 tasks and ranks second on the remaining 3, consistently outperforming prior methods including GerNA-Bind.

Notably, RiboSphere demonstrates strong robustness under out-of-distribution (OOD) settings. On the most challenging Biosensor split (Homol. & fingerpr.), RiboSphere achieves an AUROC of 0.7534, exceeding GerNA-Bind by 2.6%. We attribute this advantage to the discrete bottleneck's denoising effect: codebook quantization filters conformational noise irrelevant to binding function, retaining only discriminative structural features such as binding-pocket geometries. This mechanism enables effective generalization to novel RNA folds unseen during training, suggesting that the learned discrete vocabulary captures transferable structural primitives rather than dataset-specific patterns.

### 4.5. VQ-Structure Mapping

We further analyzed the correspondence between discrete structural tokens obtained via VQ quantization and specific RNA motifs, aiming to answer a more fundamental question: do the discrete tokens learned by the model correspond to RNA local structural units with clear biophysical meaning? We validated this from two perspectives.

First, we examined the structural consistency of frequently occurring local VQ tokens. We first identified the most frequent 5-mer and 7-mer VQ token sequences across the dataset and traced all their geometric instances in the original RNA 3D structures. As shown in Figure 3, structural segments mapped to the same discrete token sequences exhibit high spatial overlap, with average RMSD values remaining low (below 0.5 Å). At the structural level, some

5-mer tokens consistently correspond to specific types of backbone bends or torsional conformations, while certain 7-mer tokens capture more complete local structural units, such as the edges of hairpin loops or the stem–loop transition regions.

Second, starting from biologically defined RNA motifs, we analyzed their statistical signatures in the VQ token space. As illustrated in Figure 4, different motif types show clear deviations from the background token distribution. Internal loops (IL) and hairpin loops (HL) exhibit moderate KL divergence values of 0.11 and 0.12, respectively, indicating that their token usage is partially constrained while still retaining substantial geometric variability. In contrast, three-way junctions (J3) display a much higher KL divergence (0.70) and strong enrichment on a small subset of tokens, reflecting their highly constrained spatial configurations.

Furthermore, tokens across different motif types are well separated in terms of Jensen–Shannon distance, with the distances between J3 and other motifs being significantly larger than that between IL and HL. These results suggest that, although individual motif instances do not map to identical discrete token sequences, VQ tokens capture motif-specific local structural characteristics at a statistical level.

## 5. Conclusion

In this work, we introduced RiboSphere, a generative framework based on flow matching designed specifically for the reconstruction of complex RNA structures. Our empirical evaluations demonstrate that RiboSphere achieves state-of-the-art performance in structure modeling. Beyond reconstruction, we established that RiboSphere serves as a robust foundation for broader structural understanding tasks, including inverse folding and RNA-ligand binding affinity prediction. Moving forward, we aim to extend our structural representations to accommodate a wider array of downstream tasks, such as zero-shot function annotation and the modeling of dynamic RNA-protein complexes.

## Acknowledgement

Zhou Zhang and Tianfan Fu are supported by Young Scientists Fund (C Class) of the National Natural Science Foundation of China (Grant No. 62506154), the Fundamental Research Funds for the Central Universities and Nanjing University International Collaboration Initiative (Grant No. 020214380129) and the "111 Center" (No. B26023). The work described in this paper was also partially supported by the Research Grants Council of the Hong Kong Special Administrative Region, China, under Project T45-401/22-N.

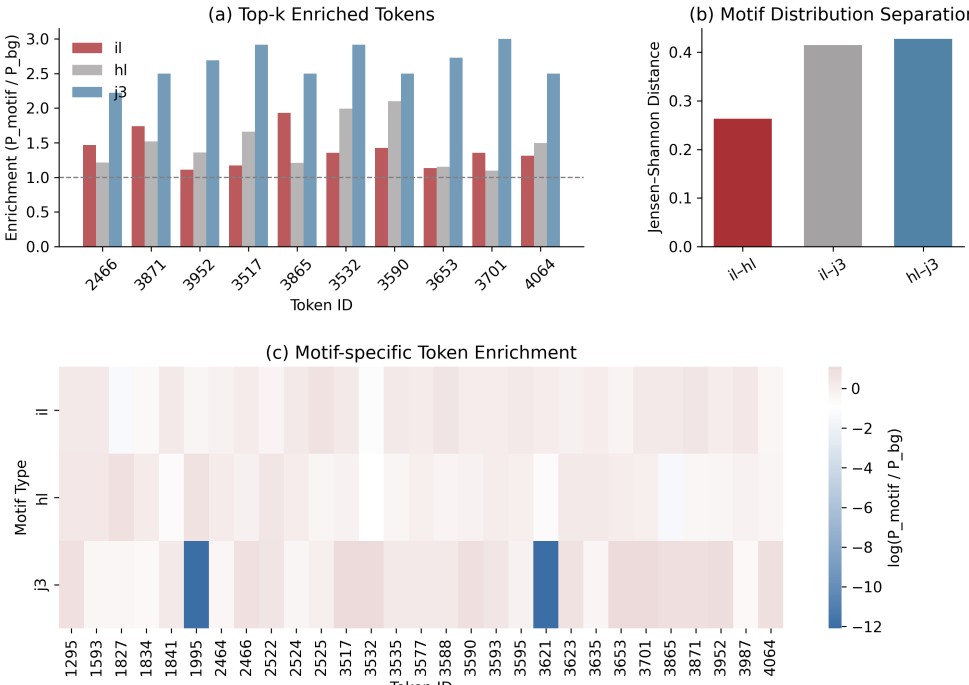

*Figure 4.* Motif-Specific Token Distributions in the VQ Structural Space

## Impact Statement

This work proposes a discrete geometric representation for RNA structure modeling, which improves the ability to model RNA structures and functions under data-scarce conditions and has the potential to advance research in RNA design, drug discovery, and synthetic biology. By learning a biophysically interpretable structural codebook, the method offers a new perspective for modular modeling of complex biological macromolecules. However, without experimental validation, the model's predictions may be biologically implausible or misleading if over-interpreted. Overall, this work is expected to contribute positively to methodological advances in computational biology, without posing direct or significant societal risks.

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

# A. Metric

This section describes all metrics used to assess model performance across the different tasks studied in this paper, together with their formal definitions and computation procedures.

## A.1. Reconstruction Metrics

For the structure reconstruction task, model performance is primarily evaluated from the perspective of three-dimensional geometric similarity. We report the following complementary metrics.

**Root Mean Square Deviation (RMSD).** RMSD measures the average deviation between predicted and ground-truth atomic coordinates in three-dimensional space. Given predicted coordinates $\hat{x}_i \in \mathbb{R}^3$ and ground-truth coordinates $x_i \in \mathbb{R}^3$ for $i = 1, \ldots, N$, RMSD is defined as

$$\text{RMSD} = \sqrt{\frac{1}{N} \sum_{i=1}^{N} ||\hat{x}_i - x_i||_2^2}$$

Prior to computation, predicted and reference structures are optimally aligned using a rigid-body superposition (Kabsch algorithm) to remove global translation and rotation. RMSD is sensitive to both local and global geometric errors and directly reflects reconstruction accuracy.

**TM-score.** ([Zhang & Skolnick, 2004](#)) TM-score evaluates the similarity of global structural topology while being relatively insensitive to structure length. It is defined as

$$\text{TM} = \frac{1}{L} \sum_{i=1}^{L} \frac{1}{1 + \left( \frac{d_i}{d_0(L)} \right)^2}, \quad d_0(L) = 1.24 \sqrt[3]{L - 15} - 1.8.$$

where $L$ denotes the structure length, $d_i$ is the distance between the $i$-th pair of aligned residues, and $d_0(L)$ is a length-dependent normalization constant. TM-score takes values in $(0, \ 1]$, with higher values indicating greater global structural similarity. Compared to RMSD, TM-score emphasizes correct overall folding rather than fine-grained local deviations.

**Local Distance Difference Test (lDDT).** lDDT assesses the accuracy of local geometric relationships without requiring global structure alignment. The metric compares pairwise distances between atoms (or residues) in the predicted and reference structures.

For all atom pairs $(i, \ j)$ whose true distance $d_{ij}$ is below a fixed cutoff (typically 15 Å), the absolute distance difference $|\hat{d}_{ij} - d_{ij}|$ is evaluated against multiple tolerance thresholds (e.g., 0.5 Å, 1 Å, 2 Å, and 4 Å). The lDDT score is computed as the average fraction of atom pairs satisfying these thresholds, yielding values in $[0, \ 1]$.

lDDT is particularly sensitive to local structural correctness and complements global metrics such as RMSD and TM-score.

## A.2. Inverse Folding Metrics

In the inverse folding task, the model generates RNA sequences conditioned on a given target structure. Performance is evaluated using sequence-level metrics.

**Sequence Recovery.** ([Dauparas et al., 2022](#)) Sequence recovery measures the fraction of positions at which the predicted sequence matches the ground-truth sequence exactly. It is defined as

$$\text{Rec} = \frac{1}{L} \sum_{i=1}^{L} \mathbb{1}(\hat{s}_i = s_i).$$

where $s_i$ and $\hat{s}_i$ denote the ground-truth and predicted nucleotide identities at position $i$, respectively, and $\mathbb{1}(\cdot)$ is the indicator function. This metric reflects the model's ability to reproduce native sequence preferences under structural constraints.

**3-mer Diversity.** Given $n$ candidate sequences designed for the same RNA backbone, we characterize sequence diversity using 3-mer frequency statistics. For each candidate sequence $s$, let $v_s \in \mathbb{R}^{64}$ denote the normalized frequency vector over all possible nucleotide 3-mers. We define the 3-mer diversity as

$$\text{Div}_{3\text{mer}} = 1 - \mathbb{E}_{s \neq t}[\rho(v_s, v_t)],$$

where $\rho(\cdot)$ denotes the Pearson correlation coefficient between two 3-mer frequency vectors. This metric quantifies sequence-level diversity by measuring the average dissimilarity of local substring usage across generated candidates. Higher values indicate greater diversity and reduced mode collapse.

### A.3. RNA–Ligand Binding Metrics.

For the RNA–ligand binding task, we formulate the problem as binary classification, predicting whether a given RNA–ligand pair is binding or non-binding. Model performance is evaluated using the Area Under the Receiver Operating Characteristic Curve (AUROC).

AUROC is defined as the area under the receiver operating characteristic (ROC) curve, which plots the true positive rate (TPR) against the false positive rate (FPR) across all possible decision thresholds. Higher AUROC values indicate stronger binding specificity and reflect threshold-independent classification performance.

## B. Implementation Details

This section provides details of the RiboSphere architecture, training pipeline, and inference procedure.

### B.1. Hyperparameters

We train all models using AdamW with $\beta_1 = 0.9$, $\beta_2 = 0.95$. Unless otherwise specified, models are trained on 8 NVIDIA A6000 GPUs using data-parallel training. Gradient accumulation is applied to achieve a stable effective batch size across devices. Each micro-step processes a single RNA structure with randomly augmented rigid-body transformations. This design avoids padding or masking across variable-length sequences and simplifies batching during training.

*Table 4.* Training configuration hyperparameters.

| Parameter | Value |
|---|---|
| Encoder layers | **2** / 4 / 6 / 8 |
| Decoder layers | 2 / 4 / 6 / **8** |
| Attention heads | 8 |
| Encoder channels | **256** / 512 |
| Decoder channels | 512 |
| Pair-bias channels | 64 |
| FSQ levels | (8, 6, 5) / (8, 5, 5, 5) / (7, 5, 5, 5, 5) |
| MLP factor | 4 |
| Learning rate | $3 \times 10^{-4}$ |
| epoch | 100 |
| Sliding window size | **8** / 16 / None |
| Gradient accumulation (micro-steps) | 8 |
| QK normalization | True / **False** |

### B.2. Data Preprocessing and Augmentation

During training, all atomic coordinates are mean-centered to remove global translations, i.e.,

$$\tilde{x} \;=\; x - \frac{1}{N}\sum_{i=1}^{N} x_i,$$

Since diffusion and flow-based generative processes cannot be defined in a translation-invariant manner over the full Euclidean space, we instead operate in the zero center-of-mass subspace. Concretely, if both the data sample $x$ and Gaussian noise $\epsilon \sim \mathcal{N}(0, \mathbf{I})$ satisfy $\sum_i x_i = \sum_i \epsilon_i = 0$, the noise interpolation used in flow matching,

$$x_t \;=\; tx + (1-t)\epsilon,$$

remains well-defined and closed under linear combinations.

To enforce rotational equivariance and improve generalization, we augment RNA structures with random 3D rotations during training. For each RNA chain, multiple rotated copies are generated by sampling rotation matrices $R$ uniformly from SO(3) and applying them to the atomic coordinates as $x' = Rx$. This augmentation strategy preserves internal geometry while preventing the model from overfitting to specific global orientations.

Additional preprocessing steps include masking invalid or missing atoms, unit normalization, and optional selection of different atomic subsets, depending on the experimental setting.

### B.3. FSQ Details

We implement the discrete bottleneck with finite scalar quantization (FSQ). Given an encoder output $c_i \in \mathbb{R}^d$ for residue $i$, we first project it to the FSQ dimension and bound each channel with a $\mathrm{tanh}$ nonlinearity. For the $k$-th FSQ channel with $L_k$ quantization levels, the bounded scalar is rounded to the nearest integer:

$$\hat{c}_{i,k} = \mathrm{round}\left(\left\lfloor \frac{L_k}{2} \right\rfloor \tanh(W_k c_i)\right),$$

where gradients through the rounding operation are propagated using the straight-through estimator. The per-channel level configuration is denoted by $\mathbf{L} = [L_1, \ldots, L_d]$, and defines an implicit Cartesian-product codebook

$$\mathcal{C} = \prod_{k=1}^{d} \{0, \ldots, L_k - 1\}, \qquad |\mathcal{C}| = \prod_{k=1}^{d} L_k.$$

Thus, each quantized vector $\hat{c}_i$ corresponds to one discrete structural token. We convert $\hat{c}_i$ to a token index by mixed-radix enumeration:

$$\mathrm{idx}(\hat{c}_i) = \sum_{k=1}^{d} \tilde{c}_{i,k} \prod_{m<k} L_m,$$

where $\tilde{c}_{i,k}$ denotes the shifted non-negative integer value of the $k$-th quantized coordinate. This gives a bijection between FSQ code vectors and integer token IDs used by the decoder and downstream modules.

In our experiments, different target vocabulary sizes are obtained by choosing different level configurations. Specifically, we use $\mathbf{L} = (8, 6, 5)$ for a 240-token codebook, $\mathbf{L} = (8, 5, 5, 5)$ for a 1000-token codebook, and $\mathbf{L} = (7, 5, 5, 5, 5)$ for a 4375-token codebook. Following the FSQ design, we keep each per-channel level count at least five when possible, which provides a compact low-dimensional quantizer while avoiding very coarse binary-style partitions.

### B.4. Flow Matching Inference Details

At inference time, we generate RNA atomic coordinates by numerically integrating the learned conditional flow from an isotropic Gaussian prior to the data distribution. We adopt classifier-free guidance (CFG), score annealing, and noise annealing, following standard practices in diffusion-based generative modeling. Unless otherwise stated, all inference hyperparameters are inherited from ProteinFlow-style settings and are not extensively tuned, as the diffusion autoencoder framework allows the noise and score scales to be adjusted flexibly for different downstream tasks.

Concretely, given a sequence of discrete structural tokens, we initialize coordinates with zero-mean Gaussian noise and evolve them using an Euler discretization of the conditional flow matching dynamics. At each time step, we evaluate both

the conditional vector field and an unconditional counterpart obtained by masking the codebook conditioning, and combine them using a classifier-free guidance weight. For intermediate time steps, stochasticity is injected through a Gaussian noise term whose variance is modulated by a time-dependent schedule, while near the terminal time the dynamics become fully deterministic.

### B.5. Inverse Folding Adapter Details

The Adapter module is designed to map the latent discrete structural representations learned by RiboSphere to sequence generation tasks. It employs a dual-path architecture that maintains both scalar and vector features: the scalar path encodes rotation- and translation-invariant chemical environment information, while the vector path explicitly models three-dimensional spatial directions to capture structural topology. Given latent node features $\hat{c}$, they are first linearly projected into scalar and vector spaces:

$$node_s = W_s\hat{c}, \quad v_{gate} = \tanh(W_v\hat{c})$$

To enable selective modulation of geometric directions, a scalar-to-vector feedback is applied:

$$v_{weight} = v_{gate} + W_{s2v}(node_s),$$

which gates local directional vectors $v_{local}$ to form updated node vectors:

$$node_v = v_{weight} \odot \text{mean}_V(v_{local})$$

To capture pairwise geometric dependencies, a pairwise message-passing operator is used. Relative positional biases generate a pairwise relation matrix $S$, which is processed via a multi-layer perceptron and a causal mask $\mathcal{M}$:

$$v_{pair} = \sum_j (\sigma(S) \odot \mathcal{M})_{ij} v_{local}^j,$$

followed by aggregation:

$$node_v \leftarrow node_v + \text{mean}_V(v_{pair}).$$

Finally, vector-to-scalar feedback projects directional information back to the scalar path:

$$node_s \leftarrow node_s + W_{v2s}(node_v),$$

enhancing the scalar features' sensitivity to spatial topology. This bidirectional interaction ensures deep integration of scalar and vector modalities, providing the subsequent flow-matching transformer with an initial state that combines semantic richness with geometric precision, guaranteeing physically plausible and spatially coherent generated structures.

## C. Additional Experimental Results

### C.1. Extended Reconstruction Results

The table 5 presents extended reconstruction results for RiboSphere, complementing the main results reported in the text. These additional experiments provide further insights into the impact of different encoder/decoder configurations, embedding dimensions, and number of atoms on structural reconstruction quality and vector quantization usage.

### C.2. Additional Inverse Folding Analysis

To further characterize the behavior of our inverse folding model, we examined the trade-off between sequence recovery and diversity under different sampling temperatures.

*Table 5.* Extended Reconstruction Results of RiboSphere.

| Method | # Enc | # Dec | Dim | # Atoms | Structure | | | VQ | |
|---|---|---|---|---|---|---|---|---|---|
| | | | | | RMSD | TM-score | lDDT | Codebook | % Util. |
| E6-D6, D512, A1 | 6 | 6 | 512 | 1 | 3.03 | 0.63 | 0.70 | 240 | 100 |
| E6-D6, D512, A10 | 6 | 6 | 512 | 10 | 2.70 | 0.67 | 0.71 | 240 | 100 |
| E6-D6, D512, A11 | 6 | 6 | 512 | 11 | 2.89 | 0.65 | 0.71 | 240 | 100 |
| E6-D6, D512, A1 | 6 | 6 | 512 | 1 | 1.88 | 0.73 | 0.75 | 1,000 | 95.0 |
| E6-D6, D512, A10 | 6 | 6 | 512 | 10 | 2.44 | 0.67 | 0.71 | 1,000 | 88.5 |
| E6-D6, D512, A11 | 6 | 6 | 512 | 11 | 2.71 | 0.68 | 0.72 | 1,000 | 98.9 |
| E6-D6, D512, A1 | 6 | 6 | 512 | 1 | 2.16 | 0.73 | 0.76 | 4,375 | 39.2 |
| E6-D6, D512, A10 | 6 | 6 | 512 | 10 | 2.01 | 0.74 | 0.75 | 4,375 | 46.6 |
| E6-D6, D512, A11 | 6 | 6 | 512 | 11 | 2.34 | 0.68 | 0.71 | 4,375 | 35.6 |
| E4-D6, D256, A11 | 4 | 6 | 256 | 11 | 2.43 | 0.67 | 0.72 | 240 | 100 |
| E4-D6, D512, A11 | 4 | 6 | 512 | 11 | 2.46 | 0.67 | 0.71 | 240 | 100 |
| E4-D6, D256, A11 | 4 | 6 | 256 | 11 | 2.58 | 0.66 | 0.71 | 1,000 | 81.2 |
| E4-D6, D512, A11 | 4 | 6 | 512 | 11 | 3.10 | 0.64 | 0.70 | 1,000 | 80.5 |
| E4-D6, D256, A11 | 4 | 6 | 256 | 11 | 2.70 | 0.71 | 0.74 | 4,375 | 39.8 |
| E4-D6, D512, A11 | 4 | 6 | 512 | 11 | 2.13 | 0.72 | 0.75 | 4.375 | 38.6 |

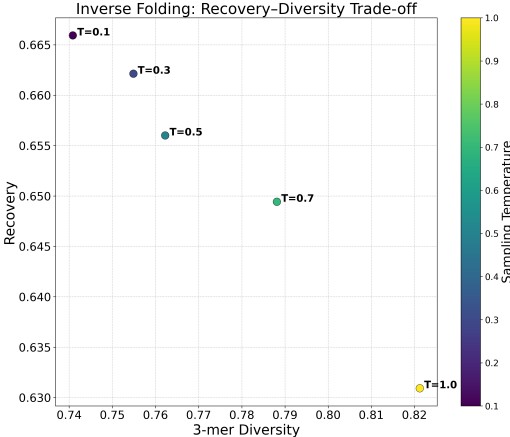

*Figure 5.* Recovery–Diversity trade-off for the inverse folding model under different sampling temperatures.

Figure 5 shows the Recovery–Diversity trade-off curve obtained from sampling 16 sequences per structure at temperatures $T = \{0.1, 0.3, 0.5, 0.7, 1.0\}$. As expected, lower temperatures yield higher recovery rates, with $T = 0.1$ achieving the highest mean recovery of $0.666$. Conversely, higher temperatures promote sequence diversity, with $T = 1.0$ producing the largest 3-mer diversity of $0.821$. The observed trend demonstrates the inherent trade-off: improving sequence recovery generally reduces diversity, and vice versa. Intermediate temperatures around $T = 0.5$–$0.7$ provide a balanced compromise between recovery and diversity.

Overall, this analysis highlights the flexibility of our model in generating multiple candidate sequences while controlling the trade-off via sampling temperature. This capability is particularly useful for downstream RNA design applications where both fidelity to the native structure and sequence variability are desirable.

