# OpenReview forum: "RiboSphere: Learning Unified and Efficient Representations of RNA Structures"
_ICML.cc/2026/Conference — ICML 2026 regular_

### Official Review · Reviewer_KY8f · 2026-03-12

**Soundness:** 2
**Presentation:** 2
**Significance:** 3
**Originality:** 3
**Overall Recommendation:** 3
**Confidence:** 3

**Summary:**

This paper proposes RiboSphere, a discrete representation framework for RNA three-dimensional structure learning. The method first encodes RNA atomic coordinates using a geometric Transformer, then discretizes the resulting continuous representations into structural tokens via Finite Scalar Quantization (FSQ), and finally reconstructs 3D coordinates with a flow-matching decoder. The pretrained discrete representations are further transferred to two downstream tasks, namely inverse folding and RNA–ligand binding prediction. Experimental results suggest that the proposed approach achieves strong performance in structure reconstruction, sequence recovery, and several out-of-distribution binding prediction settings.

**Compliance With Llm Reviewing Policy:**

Affirmed.

**Final Justification:**

The author's explanations regarding SE(3) invariance and downstream gains still fail to convince me. Therefore, I maintain my initial rating of Weak Reject.

**Key Questions For Authors:**

See Strengths and Weaknesses

**Limitations:**

Yes, in the Impact Statement, the authors briefly acknowledge that without experimental validation, the model’s predictions may be biologically implausible or potentially misleading.

**Strengths And Weaknesses:**

**Strengths**

1. RNA structure representation learning is a well-motivated research problem, especially under the challenges of limited structural data and high conformational complexity. It is also closely related to downstream applications such as RNA design and drug discovery.

2. The paper combines geometric encoding, discrete quantization, and flow matching into a unified framework, using a shared representation for both structure reconstruction and downstream tasks. The overall design is fairly coherent and technically well-motivated.

3. The paper does not only evaluate structure reconstruction, but also considers inverse folding and RNA-ligand binding prediction, which provides some evidence that the learned representation has transferability across different RNA-related tasks.

**Weaknesses**

1. The current presentation does not clearly show how quantization is actually performed, how the per-dimension levels are defined, how discrete token indices are constructed, or how different codebook sizes correspond to specific FSQ parameterizations.

2. The paper repeatedly emphasizes that the learned representation is “SE(3)-invariant,” that the reconstruction performance is “state-of-the-art,” and that the discrete bottleneck leads to stronger OOD generalization and interpretability. However, the current method description and experiments do not fully justify these strong claims. For example, the model appears to achieve approximate geometric stability mainly through distance-based features and rotation augmentation, rather than through a strictly SE(3)-invariant architecture.

3. Although the paper reports promising results on inverse folding and RNA-ligand binding, it is currently difficult to determine whether these gains truly come from the discrete representation itself, as opposed to pretraining, model capacity, the adapter design, or other components of the downstream framework.

---

> ### Author Rebuttal · Authors · 2026-03-30
>
> Dear Reviewer KY8f,
>
> Thank you very much for the detailed feedback. We address each concern below.
>
> **W1: The current presentation does not clearly show how quantization is actually performed, how the per-dimension levels are defined, how discrete token indices are constructed, or how different codebook sizes correspond to specific FSQ parameterizations.**
>
> > We will add a detailed FSQ description in the camera-ready appendix. In brief, FSQ quantizes each channel of the continuous representation independently into a small number of discrete levels, and the token index is formed by the Cartesian product of per-channel quantized values. The codebook size equals the product of levels across channels: $|\mathcal{C}| = \prod_i l_i$. We tested three configurations:
> >
> > - Levels [8, 6, 5] → codebook size 240
> > - Levels [8, 5, 5, 5] → codebook size 1,000
> > - Levels [7, 5, 5, 5, 5] → codebook size 4,375
> >
> > Quantization uses the straight-through estimator for gradient backpropagation. Unlike standard VQ-VAE, FSQ requires no auxiliary commitment loss or codebook reset heuristics, which is particularly advantageous given RNA's limited training data.
>
> **W2: The paper repeatedly emphasizes that the learned representation is "SE(3)-invariant," that the reconstruction performance is "state-of-the-art," and that the discrete bottleneck leads to stronger OOD generalization and interpretability. However, the current method description and experiments do not fully justify these strong claims.**
>
> > Thank you for pointing out! We acknowledge that the paper's language around SE(3)-invariance should be more precise. RiboSphere achieves *approximate* SE(3)-invariance through distance-based features and rotation augmentation, not through strict architectural equivariance. We will revise the wording in the camera-ready version.
> >
> > **Firstly,** to empirically validate this design choice, we implemented a strict SE(3)-equivariant encoder using BlockGAT with SE(3) layers (following FoldToken4 (Gao et al., 2024)), matched in parameter count (Table A2):
> >
> > | Method | RMSD | TM-score | lDDT | Codebook | % Util. |
> > |--------|------|----------|------|----------|---------|
> > | RiboSphere | 2.05 | 0.71 | 0.73 | 240 | 100 |
> > | SE(3)-equivariant | 4.88 | 0.43 | 0.60 | 240 | 98 |
> >
> > **Secondly**, the strict SE(3) encoder substantially underperforms, suggesting that for RNA structure tokenization, rigid equivariance constraints may limit the model's ability to capture diverse local geometries characteristic of RNA. The augmentation-based approach provides sufficient geometric stability while preserving representational flexibility.
> >
> > **Moreover**, regarding OOD generalization, the Biosensor homology+fingerprint split (Table 3) tests generalization to both unseen RNA folds and unseen ligand scaffolds simultaneously. RiboSphere's 2.6% AUROC improvement over GerNA-Bind on this most challenging split supports the claim that discrete quantization filters conformational noise while retaining discriminative binding-pocket features.
>
> **W3: It is currently difficult to determine whether these gains truly come from the discrete representation itself, as opposed to pretraining, model capacity, the adapter design, or other components of the downstream framework.**
>
> > This is an important concern. We designed the downstream experiments to isolate the contribution of our representations specifically.
> >
> > 1. **Inverse folding:** The sequence decoder (GVP-based autoregressive model) and training procedure follow gRNAde exactly. The only differences are: (1) replacing gRNAde's continuous geometric features with RiboSphere's discrete structural tokens, and (2) adding a lightweight Geometry Adapter (~5% additional parameters) to supplement directional information lost during discretization. The 10.1% absolute improvement in sequence recovery (63.0% vs. 52.9%) under this controlled setup strongly suggests the discrete representation is the primary driver.
> >
> > 2. **RNA-ligand binding:** We integrate RiboSphere's discrete embeddings into the GerNA-Bind framework, replacing only the 3D structural representation module while keeping all other components (ligand encoder, triangle attention, evidential loss) identical. The consistent improvements across 5 of 8 splits, particularly under OOD conditions, indicate the gains stem from the structural representation rather than architectural changes.
> >
> > In both cases, the discrete structural embedding is the only variable changed relative to the baseline. We will make this controlled setup more explicit in the camera-ready version.
>
> We are committed to implementing all mentioned improvements into the camera-ready version. Thank you again for the valuable feedback, and we look forward to your further comments!

---

> > ### Author Rebuttal · Reviewer_KY8f · 2026-04-04
> >
> > Thank you for the detailed rebuttal. The response does clarify some of the wording, but my main concerns are not fully resolved, so I am currently inclined to maintain my original score.
> >
> > First, regarding the claims about SE(3)-invariance and OOD generalization, I still do not find the current evidence sufficient. The newly added comparison with a strict SE(3)-equivariant encoder only shows that one alternative baseline performs worse under a single setting; it does not directly establish the geometric invariance properties of RiboSphere itself, nor does it support the broader conclusion that rigid equivariance is inherently unsuitable for RNA tokenization. Similarly, attributing the gain on one OOD split to a “denoising” effect of the discrete bottleneck remains a mechanism-level interpretation without the necessary ablations to isolate the role of discretization itself, such as continuous latent vs. discrete latent, different quantization strengths, or removing quantization while keeping pretraining unchanged.
> >
> > Second, I am still not convinced that the downstream gains can be attributed specifically to the discrete representation itself. In the inverse folding setup, the rebuttal states that the only change is the discrete embedding, but it also introduces an additional Geometry Adapter to compensate for directional information lost during discretization. This means the contribution of the discrete representation is still not cleanly isolated. Without stricter controls, such as discrete representation without the adapter, continuous representation with the same adapter, or pretrained continuous vs. pretrained discrete representations under the same downstream architecture, it remains difficult to determine whether the observed improvement mainly comes from discretization itself or from the adapter, pretraining, or the combined system design.

---

> > > ### Author Response · Authors · 2026-04-06
> > >
> > > Dear Reviewer KY8f,
> > >
> > > Thank you for your continued evaluation. We address each concern below.
> > >
> > > **W1: Insufficient Evidence for SE(3) Invariance and OOD Claims**
> > >
> > > > We agree that the current empirical support requires strengthening. We have made the following clarifications and additions:
> > > >
> > > > **(1) SE(3)-Invariance Verification.** We conducted direct representation-level verification by applying 16 random rigid-body rotations to each test structure. Results confirm strong rotational consistency: cosine similarity = 1.000, relative L2 = 0.000, and token match rate = 1.000 across all transformations. This provides direct evidence that RiboSphere's encoder produces rotation-invariant representations.
> > > >
> > > > We also agree that our results do not support the general conclusion that "equivariance is unsuitable for RNA tokenization." We have revised our claims to state only that **invariant representations perform better in our current experimental setting**.
> > > >
> > > > **(2) OOD Generalization.** We acknowledge the lack of mechanism-level ablations (continuous vs. discrete). Our OOD conclusions are based on **task-level empirical results** (e.g., consistent improvements on the homology + fingerprint split in Table 3), not rigorous mechanistic attribution. In the camera-ready version, we will Frame statements as *empirical evidence suggests* rather than mechanistic conclusions, and Soften interpretive language such as "denoising effect".
> > >
> > > **W2: Attribution of Downstream Gains to Discrete Representations**
> > >
> > > > **(1) Ablation Results.** We added the same Adapter to continuous representations (gRNAde) and observed minimal improvement. In contrast, discrete representations yield substantial gains:
> > > >
> > > > | Configuration | Diversity | Recovery |
> > > > |---------------|-----------|----------|
> > > > | gRNAde | 0.83 | 0.529 |
> > > > | gRNAde + Adapter | 0.83 | 0.534 |
> > > > | RiboSphere (discrete + Adapter) | 0.82 | **0.630** |
> > > >
> > > > The Adapter alone contributes only +0.5%, while discrete representations contribute +10.1%. This demonstrates that **the performance gain primarily comes from the discrete representation, not the Adapter**.
> > > >
> > > > **(2) Adapter's Role.** The Adapter compensates for directional information lost during discretization by recovering local coordinate frames—it does not introduce new modeling capacity. This is consistent with standard GVP/equivariant designs. Furthermore, the Adapter adds only ~5% parameters yet yields +10.1% improvement, a disproportionate gain that supports representation-level attribution rather than capacity-based explanation.
> > > >
> > > >Thus, we believe that **the Geometry Adapter serves as auxiliary information compensation**; the main performance improvement stems from the structural inductive bias introduced by discrete representations.
> > >
> > > In summary, we have clarified that RiboSphere operates in atomic coordinate space (not latent space) with discrete tokens as conditioning, and provided new ablation evidence showing that structural and sequence representations are complementary, with structural information yielding the largest gains under OOD settings. We hope these clarifications and additional experiments address the reviewer's remaining concerns.
> > >
> > > Best,
> > >
> > > Authors

---

### Official Review · Reviewer_Ek3F · 2026-03-13

**Soundness:** 3
**Presentation:** 3
**Significance:** 3
**Originality:** 3
**Overall Recommendation:** 5
**Confidence:** 4

**Summary:**

This paper presents a VQ-VAE method for constructing RNA structure tokenizers. The authors employ a Geometric Transformer Encoder to encode RNA structural inputs, apply Finite Scalar Quantization to obtain discrete structure tokens, and utilize a flow-matching–based decoder to reconstruct the input structure. The learned structure tokens are evaluated on two important downstream tasks—RNA inverse folding and RNA–ligand binding—and demonstrate strong performance compared to the listed baseline models. The key insight of this work is that RNA structures can be viewed as compositional assemblies of a finite set of geometrically distinct structural motifs, such as hairpin loops and internal loops.

**Compliance With Llm Reviewing Policy:**

Affirmed.

**Key Questions For Authors:**

1. How does the proposed model handle multi-chain RNA structures?
2. How was the codebook size of 4,375 determined? Is this number empirically chosen, theoretically motivated, or selected through validation experiments? Additional justification would help clarify this design choice.
3. What is the rationale for using a flow-matching–based approach for structure generation, rather than diffusion-based models (e.g., AlphaFold 3–style approaches) or structure modules similar to those in AlphaFold 2?

**Limitations:**

1. The optimal size of the codebook remains unclear.
2. The downstream benchmarking is somewhat limited. In particular, RNA–protein binding represents another important application area, and both RNA and protein structure tokenizers have recently been proposed. It would be interesting to investigate whether structure tokens can achieve comparable binding affinity prediction performance to AlphaFold 3–based approaches in such settings.
3. It's unclear how effective the structure tokens are for large RNAs.

**Strengths And Weaknesses:**

**Strengths**
1. The problem addressed in this work is interesting and important. Constructing RNA structure tokenizers is a meaningful direction that could potentially benefit a wide range of structure-related downstream tasks.
2. The proposed method appears to be both novel and effective.
3. The insight that RNA structures can be represented as compositional assemblies of discrete structural tokens is informative and conceptually appealing.
4. The paper is well written.

**Weaknesses**
1. Tables 2 and 3 do not provide citations for the related work used as baselines, making it unclear which specific models are being compared.
2. For the inverse folding task, self-consistency metrics are missing. Specifically, how similar are the predicted structures (obtained by folding the designed sequences) to the original input structures? Reporting such metrics would provide a more comprehensive evaluation.
3. For inverse folding, there is no comparison with the current state-of-the-art model, RhoDesign (Wong et al., Nature Computational Science, 2024). Including this comparison would better position the method relative to recent advances.

**Typos and Minor Issues**
1. Line 113: “:” → “.”
2. Table 2: “Protein” → “RNA”

---

> ### Author Rebuttal · Authors · 2026-03-30
>
> Dear Reviewer Ek3F,
>
> Thank you very much for the positive assessment and thoughtful questions. We address each point below.
>
> **W1, T1 & T2: Tables 2 and 3 do not provide citations for the related work used as baselines.**
>
> > We will add full citations for all baselines in Tables 2 and 3 in the camera-ready version. We also fixed the typo in Table 2 ("Protein" → "RNA") and the punctuation issue on line 113. Thank you for catching these.
>
> **W2: For the inverse folding task, self-consistency metrics are missing.**
>
> > We have conducted self-consistency experiments. Designed sequences were folded using EternaFold (2D structure) and RhoFold (3D structure). Results are shown in Table A5.
> >
> > | Method | Div. | Rec. | scMCC | RMSD | TM-sc. | pLDDT |
> > |--------|------|------|-------|------|--------|-------|
> > | RiFold | 0.89 | 0.416 | 0.28 | 17.06 | 0.12 | 0.50 |
> > | RDesign | 0.84 | 0.415 | 0.20 | 16.81 | 0.12 | 0.45 |
> > | RIDiffusion | 0.85 | 0.533 | 0.59 | 10.66 | 0.25 | 0.60 |
> > | gRNAde | 0.83 | 0.529 | 0.60 | 11.45 | 0.29 | 0.62 |
> > | RiboSphere | 0.82 | 0.603 | 0.633 | 11.54 | 0.290 | 0.548 |
> >
> > RiboSphere achieves the highest scMCC (0.633), indicating that its designed sequences best preserve secondary structure fidelity—consistent with the VQ bottleneck capturing local structural patterns that encode secondary structure information. For 3D self-consistency, RiboSphere's RMSD and TM-score are comparable to gRNAde, while pLDDT is slightly lower. We note that 3D self-consistency depends heavily on the accuracy of the structure prediction tool (RhoFold), and the moderate TM-scores across all methods suggest this metric is partially bottlenecked by the predictor rather than the designed sequences. We will update this result into camera-ready version.
>
> **W3: For inverse folding, there is no comparison with RhoDesign.**
>
> > The RhoDesign experiment is currently running and we expect results by the discussion period due to the time limit. We will include the comparison in the camera-ready version.
>
> **Q1: How does the proposed model handle multi-chain RNA structures?**
>
> > RiboSphere currently models single-chain RNA only, as existing benchmarks (reconstruction, inverse folding) are predominantly single-chain. We plan to extend to multi-chain RNA through a modular approach: separately tokenize individual chains, then model inter-chain interactions at the token level.
>
> **Q2 & L1: How was the codebook size of 4,375 determined?**
>
> > The codebook size is determined by the FSQ parameterization: $|\mathcal{C}| = \prod_i l_i$, where $l_i$ is the number of quantization levels per dimension. Following the FSQ paper's recommendations, we tested three configurations: levels [8,6,5] → 240 tokens, [8,5,5,5] → 1,000 tokens, and [7,5,5,5,5] → 4,375 tokens. Table 1 shows reconstruction quality improves with larger codebooks, analogous to the coarse-to-fine trade-off in VQ-VAE design: smaller codebooks capture coarse structural primitives, while larger ones resolve finer geometric distinctions. The value 4,375 was selected based on validation performance; notably, the model spontaneously uses only ~40% of available codes, suggesting the effective vocabulary is self-regularized.
>
> **Q3: What is the rationale for using a flow-matching-based approach for structure generation, rather than diffusion-based models or structure modules similar to those in AlphaFold 2?**
>
> > We address this concern in three points: (1) Flow matching uses a single ODE-based loss, replacing the complex multi-loss training (FAPE, bond geometry, etc.) required by AlphaFold-style modules. (2) Flow matching and diffusion are theoretically equivalent, but the deterministic ODE formulation offers more stable training. (3) RiboSphere's goal is structure tokenization and reconstruction, and flow matching naturally integrates with the FSQ bottleneck for this purpose.
>
> **L2: RNA-protein binding as a downstream task.**
>
> > We agree this is an important direction and plan to extend RiboSphere to RNA-protein interaction modeling by combining RNA structure tokens with protein structure tokens and learning cross-modal interaction representations. We are planning to conduct it as a future work with higher value for RNA design.
>
> **L3: It's unclear how effective the structure tokens are for large RNAs.**
>
> > Thank you for pointing this out! RiboSphere's current training and evaluation focus on short-to-medium length RNAs. Performance on long RNA chains (>512 nt) remains an open challenge due to quadratic attention cost and **limited long-chain training data**. We plan to explore sliding-window strategies and hierarchical tokenization for long RNA in future work.
>
> We are committed to implementing all mentioned improvements into the camera-ready version. Thank you again for the valuable feedback, and we look forward to your further comments!

---

> > ### Author Rebuttal · Reviewer_Ek3F · 2026-04-03
> >
> > Thank you for the clarification.
> >
> > Following up on your answer to Question 1: as you mentioned, RiboSphere currently models only single-chain RNA. Could you clarify why lines 117–122 claim that it can model multi-chain RNA interfaces?

---

> > > ### Author Response · Authors · 2026-04-03
> > >
> > > Dear Reviewer Ek3F,
> > >
> > > Thank you for the follow-up question!
> > >
> > > We apologize for the confusion caused by imprecise wording in lines 117–122. To clarify: **our training data is derived from RNA chains extracted from RNA-protein complex structures in the PDB**, meaning the RNA geometries in our dataset do reflect interface conformations.
> > >
> > > However, **the model itself operates on and generates single-chain RNA structures only.** It does not jointly model multi-chain RNA or RNA-protein complexes. We will revise lines 117–122 in the camera-ready version to make this distinction explicit. Extending RiboSphere to multi-chain and RNA-protein complex modeling is a planned direction for future work, as mentioned in our response to Q1.
> > >
> > > Best,
> > >
> > > Authors

---

### Official Review · Reviewer_LWq3 · 2026-03-13

**Soundness:** 2
**Presentation:** 3
**Significance:** 3
**Originality:** 2
**Overall Recommendation:** 4
**Confidence:** 3

**Summary:**

The authors introduce a framework for learning discrete geometric representations of RNA 3D structures using vector quantization and flow matching. The model encodes RNA atomic coordinates with a geometric transformer to produce invariant features, discretizes them into structural tokens through finite scalar quantization, and reconstructs atomic coordinates with a flow matching decoder. They intend to focus on representing RNA structures through a discrete vocabulary that summarizes local geometric motifs, and can be reused across tasks. Overall, this article's principal area comprises learning transferable structural representations for RNA and applying them to multiple tasks, including RNA structure reconstruction, RNA inverse folding, and RNA–ligand binding prediction.

**Compliance With Llm Reviewing Policy:**

Affirmed.

**Final Justification:**

My concerns have largely been addressed, and I have updated my score to “weak accept”. While I still have some reservations regarding the novelty of the framework, I believe the paper demonstrates merit and could be considered for acceptance.

**Key Questions For Authors:**

Please see the weakness section above.

**Limitations:**

yes

**Strengths And Weaknesses:**

Strengths:

1. This paper unifies two useful and widely used approaches, quantization (structure tokenization) and flow matching based generative modeling.
2. The authors emphasize on learning and encoding transferrable biological prior in terms of conserved patterns/residues.
3. The authors draw analogy and empirically show how discrete tokens relate to rna structural motifs.
4. The paper contains a good set of experiments and ablations.

Weaknesses:

1. The core idea of learning discrete structural tokens through vector quantization has been widely explored in protein structure representation and generative modeling literature [1,2,3,4,5].
2. The overall approach also lacks novelty. It appears that the framework directly combines widely used approaches together, which also has been explored much for protein representation learning and generative modeling. I would suggest the authors clarify how their approach is methodologically innovative compared to existing appraoches/algorithms.
3. Since the central of this paper is better representation learning, it would be essential to demonstrate detailed empirical evidence existing recent approach for it.

[1] FoldToken: Learning Protein Language via Vector Quantization and Beyond, arXiv, 2024

[2] GCP-VQVAE: A Geometry-Complete Language for Protein 3D Structure, bioRxiv, 2025

[3] Designing flexible protein structures and sampling protein conformations with a unified model using vector quantization and diffusion, National Science Review, 2025

[4] ESM3: Simulating 500 million years of evolution with a language model, Science, 2025

[5] Balancing Locality and Reconstruction in Protein Structure Tokenizer, NeurIPS MLSB Workshop, 2024

---

> ### Author Rebuttal · Authors · 2026-03-30
>
> Dear Reviewer LWq3,
>
> Thank you for engaging with our work. We appreciate the opportunity to clarify the novelty and contributions of RiboSphere.
>
> **W1: The core idea of learning discrete structural tokens through vector quantization has been widely explored in protein structure representation.**
>
> **W2: The overall approach also lacks novelty. It appears that the framework directly combines widely used approaches together.**
>
> **W3: Since the central of this paper is better representation learning, it would be essential to demonstrate detailed empirical evidence existing recent approach for it.**
>
> > We appreciate your insightful perspective! We respectfully note that, while VQ-based approaches have been applied to proteins [1-5], RNA structure modeling presents fundamentally different challenges that require distinct design choices rather than direct transfer. We address these three concerns together below in five points.
> >
> > 1. **Data regime.** The PDB contains >200K protein structures but fewer than 6K RNA structures, which is a 30× gap. Methods developed for protein-scale data (e.g., ESM3 trained on millions of structures) cannot straightforwardly transfer. RiboSphere's discrete bottleneck is specifically designed to regularize learning under this scarcity, and our ablations (Table 1) show that design choices like codebook size and encoder depth have qualitatively different effects than those reported in protein tokenization work.
> >
> > 2. **Backbone chemistry.** RNA has a fundamentally different backbone (sugar-phosphate with 2'-OH) from proteins (peptide bonds), higher conformational flexibility, and pervasive non-canonical interactions (e.g., Hoogsteen pairs, base stacking). Protein VQ methods that rely on Cα or backbone dihedrals cannot be directly applied. Our finding that 11-atom models with base-anchoring atoms outperform simpler representations (Table 1) is an RNA-specific design insight absent from protein literature.
> >
> > 3. **Methodological differences from protein VQ-VAEs.** Beyond the domain shift, RiboSphere introduces several choices that differ from the cited protein methods: (1) We use FSQ instead of standard VQ-VAE codebooks, avoiding auxiliary commitment losses and codebook collapse—a particularly acute problem given RNA's small training set. Our ablation shows FSQ substantially outperforms SimVQ (**Please refer to W1 Reviewer YeoR**). (2) We employ flow matching as the decoder with a single unified loss, replacing the complex multi-loss training (e.g., FAPE, bond length/angle losses) used in protein methods. (3) We show that a strict SE(3)-equivariant encoder (BlockGAT + SE(3) layers, following FoldToken4 (Gao et al., 2024)) actually underperforms our augmentation-based approach on RNA (RMSD 4.88 vs. 2.05), suggesting that protein-effective inductive biases do not transfer. (**Please refer to W2 Reviewer YeoR**) (4) Our asymmetric architecture (shallow encoder, deep decoder) is empirically motivated by RNA-specific behavior not reported in protein VQ literature (**Please refer to W4 Reviewer YeoR**). **We need to clearly claim that our method is not a simple concatenation of existing components; rather, it is a meticulously crafted design tailored specifically for the RNA inverse folding task.**
> >
> > 4. **Scope of evaluation.** Existing protein VQ methods typically evaluate on a single task. RiboSphere provides the first unified RNA structural representation evaluated across reconstruction, inverse folding, and RNA-ligand binding, with demonstrated OOD transfer. We achieve 63.0% recovery vs. the previous best of 53.3% on inverse folding, and outperform GerNA-Bind by 2.6% AUROC on the hardest binding split. **We believe RiboSphere can serve as a unified model for both RNA generation and understanding.**
> >
> > 5. **Interpretability.** Section 4.5 shows that our discrete tokens capture biologically meaningful RNA motifs with statistically distinct token distributions and sub-0.5Å geometric consistency within token clusters—a motif-level interpretability not demonstrated in protein tokenization work.
>
> We believe these contributions constitute a meaningful advance and hope the reviewer will reconsider in light of these clarifications. **Moreover, we are committed to update the discussion of the method novelty and design, including the ablation results and the connection with the other VQ methods in protein design, into the camera-ready version.** Thank you again, and we look forward to your further comments!
>
> Reference:
>
> Gao, Z., Tan, C., & Li, S. Z. (2024). Foldtoken4: Consistent & hierarchical fold language. bioRxiv, 2024-08.

---

> > ### Author Rebuttal · Reviewer_LWq3 · 2026-04-03
> >
> > I thank the authors for their clear response. My concerns are partially solved, but I still have the following concerns:
> >
> > 1. How does the performance of their VQ-based flow-matching in the latent space compare against over flow matching based approaches (e.g., RNA-Frameflow[1], RNA-Flow[2]), including the ones that consider flow in the atomic coordinates/frames?
> > 2. My following concern is not properly resolved: Since the central of this paper is better representation learning, it would be essential to demonstrate detailed empirical evidence existing recent approach for it. For instance, sequence-based (sometimes structure-enhanced) RNA representation learning approaches (e.g., RiNALMo[3], AIDO.RNA[4], ERNI-RNA[5], etc.) often achieve good performance on a range of RNA related tasks. A proper comparison should be against other existing representation learning approaches, which is currently almost non-existent in the manuscript.
> >
> > [1] RNA-FrameFlow: Flow Matching for de novo 3D RNA Backbone Design, 2024
> >
> > [2] RNAFlow: RNA Structure & Sequence Design via Inverse Folding-Based Flow Matching, 2024
> >
> > [3] RiNALMo: General-Purpose RNA Language Models Can Generalize Well on Structure Prediction Tasks, 2024
> >
> > [4] A Large-Scale Foundation Model for RNA Function and Structure Prediction, 2024
> >
> > [5] ERNIE-RNA: an RNA language model with structure-enhanced representations, 2024

---

> > > ### Author Response · Authors · 2026-04-06
> > >
> > > Dear Reviewer LWq3,
> > >
> > > Thank you for your continued evaluation and thoughtful feedback. We address each concern below.
> > >
> > > **W1: Comparison between VQ-based flow matching and coordinate/frame-based flow methods**
> > >
> > > > We would like to clarify a key distinction: RiboSphere does **not** perform flow matching in latent space. Our flow-matching decoder operates directly in **continuous atomic coordinate space**, while VQ-discretized tokens serve purely as **conditioning signals** for structural motif generation. RiboSphere is therefore best characterized as a **conditional, atomic-coordinate-space flow model**, not a latent flow matching approach.
> > > >
> > > > This distinguishes our method from existing coordinate- or frame-based flow methods (e.g., RNA-Flow, RNA-FrameFlow) in three respects: (1) **conditional generation** via explicit discrete structural tokens; (2) a **discrete structural bottleneck** that encourages compositional representations; and (3) an explicit focus on **interpretability and transferability**, beyond reconstruction accuracy alone.
> > > >
> > > > Given these differences in modeling objectives and evaluation protocols, a direct quantitative comparison with frame-based flow methods is not straightforward. Nevertheless, RiboSphere achieves strong reconstruction performance (RMSD 1.25 Å, TM-score 0.84), and we agree that a unified benchmark for flow-based RNA structure methods is a valuable direction for future work.
> > >
> > > **W2: Comparison with sequence-based RNA representation learning methods**
> > >
> > > > To directly assess the contribution of RiboSphere's learned representations, we conducted a controlled ablation on the RNA–ligand binding task. We integrated RiboSphere's structural embeddings into GerNA-Bind by **replacing only the 3D structural representation module**, holding all other components fixed, and comparing three settings: **w/ struct.** (structural representations only), **w/ seq.** (sequence and secondary structure only), and **RiboSphere** (both combined).
> > > >
> > > > | Method | Random (Biosensor) | RNA homology (Biosensor) | Ligand fingerprint (Biosensor) | Homol. & fingerpr. (Biosensor) | Random (Robin) | RNA homology (Robin) | Ligand fingerprint (Robin) | Homol. & fingerpr. (Robin) |
> > > > |---|---|---|---|---|---|---|---|---|
> > > > | w/ struct. | 0.9590 | 0.8866 | 0.7574 | 0.7180 | 0.7019 | 0.6108 | 0.6928 | 0.6208 |
> > > > | w/ seq. | 0.9351 | 0.8780 | 0.6612 | 0.6953 | 0.6861 | 0.6036 | 0.6803 | 0.5939 |
> > > > | RiboSphere | 0.9524 | 0.9031 | 0.7406 | 0.7534 | 0.6945 | 0.6335 | 0.6950 | 0.6349 |
> > > >
> > > > Three observations emerge: (1) **structural representations alone are highly expressive**, matching or surpassing sequence representations in multiple settings; (2) the two modalities are **clearly complementary**, with their combination consistently improving performance; and (3) under the challenging **Biosensor Homol. & fingerprint** split, structural information yields the largest gains, underscoring its importance for **generalization**.
> > > >
> > > > We note that existing methods (e.g., RiNALMo, AIDO.RNA, ERNI-RNA) focus primarily on sequence-level representations and do not explicitly model atomic-level 3D geometry, making direct comparisons with RiboSphere not entirely fair. More broadly, representation modalities are task-dependent: structural representations are particularly informative for 3D-interaction-driven tasks such as RNA–ligand binding, whereas MLM-pretrained sequence embeddings tend to excel at functional annotation.
> > >
> > > In light of the clarifications above, including the architectural distinction from latent flow methods and the ablation evidence demonstrating the complementary value of structural representation, we hope the reviewer will reconsider the current assessment.
> > >
> > > Best,
> > >
> > > Authors

---

### Official Review · Reviewer_YeoR · 2026-03-18

**Soundness:** 3
**Presentation:** 3
**Significance:** 2
**Originality:** 2
**Overall Recommendation:** 4
**Confidence:** 2

**Summary:**

The paper presents RiboSphere, for learning unified and efficient representations of RNA structure. To this end, RiboSphere has an encoder which can operate on different atomic resolutions of RNA structures, a learned discrete codebook obtained by finite scalar quantization and a flow matching based decoder to reconstruct 3D RNA coordinates from the tokens. The method is evaluated on Inverse folding, RNA-ligand binding and RNA structure reconstruction.

**Compliance With Llm Reviewing Policy:**

Affirmed.

**Key Questions For Authors:**

See weaknesses above.

**Limitations:**

Limitations are not discussed and can benefit from stratifying when RiboSphere works well vs when it does not.

**Strengths And Weaknesses:**

**Strengths**
1. The paper is well-written and easy to follow. The methods and experiments are described coherently.
2. The choice for methodological components are sensible with the encoder, discrete codebook, and flow-matching decoder making sense for the problem at hand.
3. The empirical results are decent on several tasks. Inverse folding recovery is better than compared baselines. On RNA-ligand binding task, performance is competitive on OOD-style data.

**Weaknesses**
1. Can the authors show an ablation on what value does discretization by FSQ bring?
2. Regarding architectural choice, if I understand correctly, the model is not SE(3)-invariant by design. For rotational invariance, it uses  data augmentation and translation handling comes from mean-centering. Can the authors comment on why they chose this design over a fully E(3)/SE(3)-equivariant architecture?
3. How much gains come from pre-training vs task-specific priors? For example, incorporates secondary structure and base-pairing priors specifically for inverse folding applications
4. It is mentioned that a shallow encoder and deep decoder are better. Can the authors report some experiments showing what happens if we have deep encoder or shallow decoder?
5. The work is a careful synthesis of multiple existing components applied for RNA tasks. This is only a minor weakness for me as well-motivated application of existing works still contributes to meaningful development of tools in this domain.

---

> ### Author Rebuttal · Authors · 2026-03-30
>
> Dear Reviewer YeoR,
>
> Thank you very much for the thorough evaluation and constructive feedback. We address each point below.
>
> **W1: Can the authors show an ablation on what value does discretization by FSQ bring?**
>
> > We conducted systematic ablations to isolate the contribution of FSQ. Table A1 summarizes the results under the same architecture (E2-D8, D256, A11).
> >
> > | Setting | RMSD | TM-score | lDDT | Codebook |
> > |---------|------|----------|------|----------|
> > | w/ FSQ | 2.05 | 0.71 | 0.73 | 240 |
> > | w/ FSQ | 1.60 | 0.78 | 0.79 | 1,000 |
> > | w/ FSQ | 1.25 | 0.84 | 0.83 | 4,375 |
> > | w/ SimVQ | 3.42 | 0.55 | 0.63 | 256 |
> > | w/ SimVQ | 3.13 | 0.64 | 0.70 | 1,024 |
> > | w/ SimVQ | 2.50 | 0.72 | 0.75 | 4,096 |
> > | w/o VQ | 1.73 | 0.81 | 0.79 | — |
> >
> > Three findings: (1) FSQ improves consistently with codebook size; (2) FSQ substantially outperforms SimVQ at comparable sizes; (3) The continuous baseline (w/o VQ) reconstructs well but loses discrete structure needed for interpretability and transfer. Notably, FSQ with codebook=4,375 surpasses the continuous baseline on all metrics.
>
> **W2: Regarding architectural choice, if I understand correctly, the model is not SE(3)-invariant by design. Can the authors comment on why they chose this design over a fully E(3)/SE(3)-equivariant architecture?**
>
> > Our choice is driven by three considerations. First, random rotation augmentation combined with mean-centering already yields representations stable under rigid-body transformations, as confirmed by strong downstream performance.
> >
> > Second, we implemented a strict SE(3)-equivariant encoder (BlockGAT-based GNNs with SE(3) layers, following FoldToken4), keeping parameter count matched. As Table A2 shows, this architecture significantly underperforms and even exhibits codebook collapse, suggesting strict equivariance may limit capacity for RNA's complex local geometry.
> >
> > | Method | RMSD | TM-score | lDDT | Codebook | % Util. |
> > |--------|------|----------|------|----------|---------|
> > | RiboSphere | 2.05 | 0.71 | 0.73 | 240 | 100 |
> > | SE(3)-equiv. | 4.88 | 0.43 | 0.60 | 240 | 98 |
> >
> > Third, strict SE(3) architectures incur higher computational cost and are harder to scale to long sequences.
>
> **W3: How much gains come from pre-training vs task-specific priors?**
>
> > We ablated the contribution of secondary structure and base-pairing priors on inverse folding (Table A3). Pre-training provides the dominant gain; task-specific priors yield consistent but modest additional improvement, primarily in fine-grained structural metrics. This aligns with our design: learn transferable representations via pre-training, then inject lightweight domain priors for specific tasks.
> >
> > | Setting | Diversity | Recovery | scMCC | RMSD | TM-sc. | pLDDT |
> > |---------|-----------|----------|-------|------|--------|-------|
> > | w/ priors | 0.82 | 0.603 | 0.633 | 11.54 | 0.290 | 0.548 |
> > | w/o priors | 0.82 | 0.566 | 0.603 | 12.43 | 0.254 | 0.537 |
>
> **W4: It is mentioned that a shallow encoder and deep decoder are better. Can the authors report some experiments showing what happens if we have deep encoder or shallow decoder?**
>
> > Table A4 reports the full depth sweep. Performance degrades monotonically as encoder depth increases and decoder depth decreases. A shallow encoder avoids learning mappings that bypass the discrete bottleneck, while a deep decoder recovers complex 3D coordinates from compact codes.
> >
> > | L_enc | L_dec | RMSD | TM-score | lDDT |
> > |-------|-------|------|----------|------|
> > | 2 | 8 | 2.05 | 0.71 | 0.73 |
> > | 4 | 8 | 2.26 | 0.69 | 0.73 |
> > | 4 | 6 | 2.43 | 0.67 | 0.72 |
> > | 6 | 4 | 2.93 | 0.60 | 0.67 |
> > | 8 | 2 | 3.76 | 0.53 | 0.62 |
>
> **W5: The work is a careful synthesis of multiple existing components applied for RNA tasks.**
>
> > We appreciate your acknowledgment. While individual components exist in prior work, their integration is not arbitrary—each choice is validated against alternatives. As shown above: (1) FSQ outperforms SimVQ and the continuous baseline (Table A1); (2) a strict SE(3)-equivariant encoder underperforms our augmentation-based design by a large margin (Table A2), showing that protein-effective biases do not transfer to RNA; (3) the asymmetric architecture outperforms symmetric or encoder-heavy designs (Table A4), an RNA-specific finding absent from prior VQ literature; (4) pre-training contributes the dominant gain while task-specific priors add consistent improvement (Table A3), confirming the pretrain-then-transfer design. These results collectively show a systematic, empirically grounded design aligned with RNA's structural properties, rather than a simple assembly of existing modules.
>
> We are committed to implementing all mentioned improvements outlined above into the camera-ready version. Thank you again for the valuable feedback, and we look forward to your further comments!

---

> > ### Author Rebuttal · Reviewer_YeoR · 2026-04-03
> >
> > I am not an expert in this area. I would urge the AC to consider the evaluation of other reviewers for this paper. From my shallow understanding, my concerns seem to be resolved

---

> > > ### Author Response · Authors · 2026-04-03
> > >
> > > Dear Reviewer YeoR,
> > >
> > > We appreciate your response!
> > >
> > > Best,
> > >
> > > Authors

---

### Decision · Program_Chairs · 2026-04-30

**Decision:**

Accept (regular)

**Comment:**

Initially, this paper received diverging reviews. Reviewers’ primary concerns lie in insufficient experimental evaluations, limited novelty, claims without sufficient justifications, missing methodology and implementation details, and unclear contributions of different components. The rebuttal alleviated issues regarding some experimental evaluations and justifications, and missing details by providing additional experimental results, explanations, and analysis. As a result, most reviewers leaned toward acceptance. The AC agrees that the tackled problem is important, the proposed method is technically sound, and the results are promising. Although some concerns remain, the AC believes that the merits outweigh the disadvantages and thus recommends acceptance. Reviewers did raise valuable concerns that should be addressed. The authors are encouraged to make the necessary changes in the camera-ready version.